# Prevalence of refractive error among *Dongarias* and use of Folding Phoropter (FoFo) as a field device enabling implementation research in this indigenous community. Tribal Odisha Eye Disease Study (TOES) Report # 13

**Debananda Padhy**[1,2], **Taraprasad Das**[1,3], **Debasmita Majhi**[1,4], **Rohit C. Khanna**[5,6,7,8], **Komal Avhad**[1], **Akhila Bihari Ota**[9], **Prachi Parimita Rout**[9], **Suryasnata Rath**[1,10]*

1 Indian Oil Centre for Rural Eye Health, Mithu Tulsi Chanrai Campus, L V Prasad Eye Institute, Bhubaneswar, India, 2 Gullapalli Pratibha Rao International Centre for Advancement of Rural Eyecare, L V Prasad Eye Institute, Hyderabad, India, 3 Anant Bajaj Retina Institute- Srimati Kanuri Santhamma Centre for Vitreoretinal Disease, L V Prasad Eye Institute, Banjara Hills, Hyderabad, India, 4 Pediatric Ophthalmology, Strabismus, and Neuro-Ophthalmology, Child Sight Institute, Miriam Hyman Children's Eye Care Centre, Mithu Tulsi Chanrai Campus, L V Prasad Eye Institute, Bhubaneswar, India, 5 Allen Foster Community Eye Health Research Centre, Gullapalli Pratibha Rao International Centre for Advancement of Rural Eye care, L V Prasad Eye Institute, Hyderabad, India, 6 Brien Holden Eye Research Centre, L.V. Prasad Eye Institute, Banjara Hills, Hyderabad, India, 7 School of Optometry and Vision Science, University of New South Wales, Sydney, Australia, 8 University of Rochester, School of Medicine and Dentistry, Rochester, NY, United States of America, 9 Scheduled Castes and Scheduled Tribes Research and Training Institute, Government of Odisha, Bhubaneswar, India, 10 Ophthalmic Plastics, Orbit, and Ocular Oncology Services, Mithu Tulsi Chanrai campus, L V Prasad Eye Institute, Bhubaneswar, India

* suryasnata@lvpei.org

## Abstract

### Purpose

To evaluate the prevalence of uncorrected refractive error (URE) among *Dongarias*—a particularly vulnerable tribal group in Rayagada, Odisha, India and evaluate if folding phoropter (FoFo) can help achieve on-site correction of URE.

### Methods

This was a cross-sectional study. FoFo was used for people with URE. Spherical equivalent (SE) spectacles based on the FoFo refraction were dispensed when distance visual acuity improved to > 6/12. Others were referred to fixed eye centres. Multivariable logistic regression evaluated the relationship of URE with sociodemographic characters and factors predicting acceptance of FoFo.

### Results

In the study, 7.5% (95% confidence interval [CI]:7–8) people had URE, and included 4% with severe vision impairment and 36% with moderate visual impairment. URE was less common in children. Simple hyperopia was more common in females (25.3% vs 19.3%);

Disease Study is the property of Scheduled Caste and Scheduled Tribe Research and Training Institute(SCSTRTI), Bhubaneswar, India. Data can be accessed only by writing to the Director, SCSTRTI at Director SCSTRTI <scstrti@yahoo.co.in>

**Funding:** Hyderabad Eye Research Foundation, L V Prasad Eye Institute, Hyderabad, India; Ministry of Tribal Affairs, Government of India; SC&ST Research Training Institute, Bhubaneswar, India; Naraindas Morbai Budhrani Trust, Mumbai, India.

**Competing interests:** The authors have declared that no competing interests exist

compound myopic astigmatism was more common in males (27.4% vs 20.2%). People older than 50 years (p <0.0001) and females (p <0.0001) were more likely to have URE. Ninety-four people accepted FoFo on-site refraction and received SE spectacles; the acceptance was better in the younger (15–29 years old) and literate people (p <0.0001).

## Conclusions

URE was the second most common cause of visual impairment in older adults and was higher in females. Within its technological limit, FoFo could be used in the field for correcting URE and obviating the need for travel, thus bridging the know-do gap for the marginalised *Dongaria* community.

## Introduction

The eastern state of Odisha in India is home to many indigenous communities. The *Dongaria* community is one of them. They reside in the mountainous terrain of Rayagada in southwest Odisha. This community is marginalised due to various factors, including poverty, poor literacy, and limited access to health care. The Government of India has included them as one of the 75 'particularly vulnerable tribal groups' (PVTG) based on population, habitat, livelihood opportunities, and literacy criteria. This community is more primitive and marginalised than the other tribal communities in India. There are 13 such PVTG communities in Odisha residing in 14 (of 30) districts, and the total population is 138 125 as per the 2011 census [1].

In association with the Ministry of Tribal Affairs, the Governments of India, and Odisha, we completed a protocol-based comprehensive eye health survey [2]. This survey identified the people with eye disorders and provided necessary treatment to reduce avoidable visual impairment in this community. Given this community's poor health-seeking behaviour and poor health compliance, every attempt was made to provide the required services as quickly as possible, at least inconvenience and at no cost to the people and the community. It also included dispensing the spectacles for refractive errors and presbyopia.

Uncorrected refractive error (URE) is the principal cause of moderate visual impairment globally [3]. Refractive error correction needs estimation of the eye's refractive status and dispensing of the required spectacles. Traditionally, the expertise of a trained ophthalmic technician/ optometrist is necessary to estimate the refractive error. Unfortunately, there is an acute shortage of trained ophthalmic personnel to meet the global need, especially in low- and middle-income countries [4]. Hence, mechanized refraction with an autorefractometer offers a reasonable alternative. A more advanced refraction system is a phoropter. It incorporates a series of lenses required for subjective correction. The autorefractometer and phoropter are often combined as a single unit for easy refraction. This bulky equipment is usually housed in an eye examination room and is difficult to carry to a remote screening site. However, technological advancements have made it more reliable and less bulky. Since dispensing spectacles soon after the refraction tends to improve compliance [5], we used an in-house designed device, the folding phoropter (popularly known, FoFo (https://lvpmitra.com ›phoropter)) for autorefraction at the screening site. It is a portable, hand-held, lightweight device with a telescopic layout that permits two lenses to slide toward or away from each other. Given that our study is the first universal population-based comprehensive eye screening in the *Dongaria* tribals, the data is unique and representative of this community like no previous study.

This paper reports the prevalence of URE among the *Dongaria* indigenous community in the Rayagada district of Odisha, India. It investigates whether the on-site use of the FoFo among the minimally literate *Dongaria* community helped correct the URE.

## Methods

The protocol for detecting a refractive error, as a part of the 'Tribal Odisha Eye Disease Study (TOES)- PVTG', has been published [2]. It was approved by the Institutional Ethics Committee, L V Prasad Eye Institute, Bhubaneswar, Odisha (2021-76-BHR-39/IEC, amended 20 August 2021) and was performed according to the tenets of the Declaration of Helsinki.

In brief, it was a population-based eye health study that included screening at the community level, comprehensive eye examination at the Vision Center (VCs), and treatment–medical/surgical- at the Secondary (Community) Center (SC). The median distance from the community screening to the three VCs was 22 km, and to SC was 60 km. The community-level eye screening consisted of measurement of distant visual acuity using a portable self-illuminated vision chart (E- optotypes, Appasamy Associates, India) placed at 3 meters distance, near vision examination by using five tumbling E optotypes corresponding to N8 at 40 centimetres and gross flashlight examination of the eye. We used the Basic Eye Screening Test (BEST) protocol [6]. But we amended the protocol to add on-site refraction of consenting individuals, given that most people were reluctant to travel to the VCs, even at no cost to them. The objective refraction was done using the FoFo. The Folding Phoropter is a foldable paper device that measures the refractive error. It consists of a converging and diverging lens (Fig 1A and 1B). The target image becomes clear by adjusting the distance between the lenses, and the spherical equivalent is captured with the help of a scale marked on the device. The spherical equivalent-based spectacles were dispensed on-site or hand-delivered at their residence within two weeks.

## Definition of types of refractive error

In this study, we used the following definitions to categorise refractive errors:

Simple myopia- a spherical equivalent of $\geq$ -0.50 D in one or both eyes,

Simple hyperopia- a spherical equivalent of $\geq$ +0.50 D in one or both eyes,

Astigmatism- a cylinder power equivalent of $\geq$ 0.50 D or more in one or both eyes.

These were divided into three categories: myopic (simple and compound), hyperopic (simple and compound), and mixed astigmatism.

Simple myopic and hyperopic astigmatism- a cylinder power of $\geq$ 0.50 D in one meridian,

Compound myopic and hyperopic astigmatism- a cylinder power of $\geq$ 0.50 D in both meridians. Mixed astigmatism- myopia in one meridian and hyperopia in the other with a cylinder error more remarkable than the sphere.

Written consent from the village head preceded the screening. For minors (participants younger than 18 years old), we obtained consent from their parents. Verbal informed consent for screening was obtained from all people in their spoken language, *Kui* (this community does not have a written language). After explaining the consent content to the study subject, a digital thumb scan was obtained. Staff at the hospital who knew the *kui* explained the procedure. *Dongarias* were informed that all examinations and treatment if any, would be done at no cost. This included the travel cost to VCs and SC, should this be required. At SC, written consent was taken before an intervention, and the left thumb impression was taken from illiterate people.

Community health workers (CHWs) recruited from the *Dongaria* community and two trained vision technicians (VTs) outside of this community collected the basic health and eye

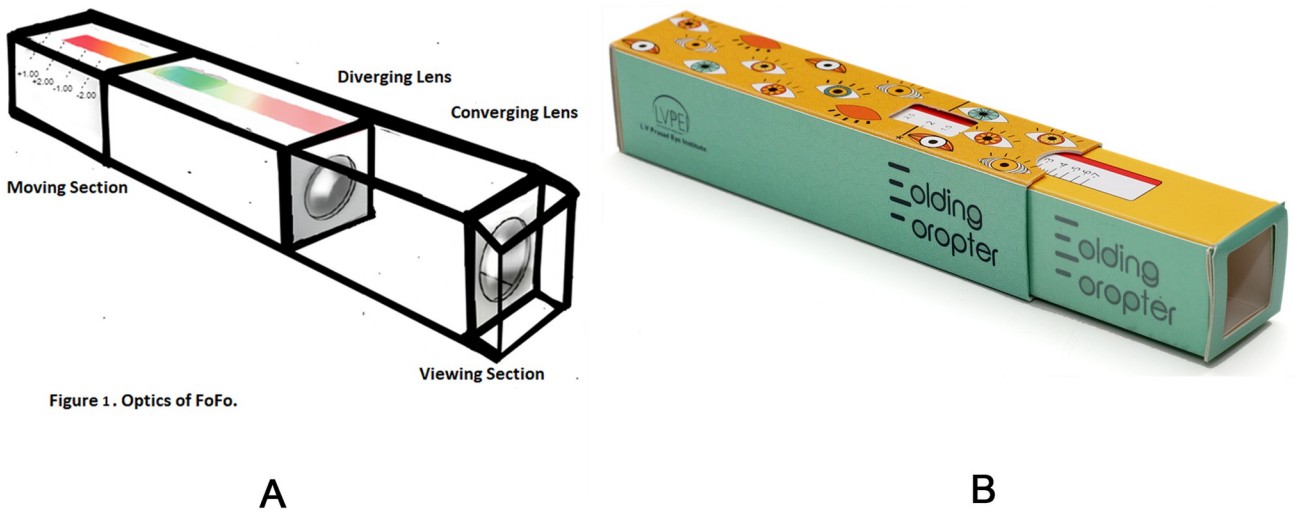

Fig 1. A: Optics of FoFo and B: showing the actual device.

health data and the demographic details through a door-to-door visit. They recorded the distance and near vision using the BEST protocol with (if worn) and without spectacles. People with distant vision 6/12 or worse received a pinhole distance vision test, and those who improved were offered to undergo on-site refraction with FoFo. People who failed to improve with pinhole were referred to fixed centres for further examination (Fig 2). At VC or SC, an experienced VT or optometrist performed a standard retinoscopy. A subjective acceptance determined the final correction. TOES-PVTG was conducted between 16th July 2021 to 30th January 2022.

Demographic details, gender, age, and literacy level were collected from each participant. Individuals were designated as illiterate when they had never attended school and could not read or write in one language. Conversely, individuals were considered literate when they had been to school and could read and write in one language.

## Statistical analysis

URE prevalence, mean, median, and range calculations were performed for age and types of refractive error. Univariate and multivariate logistic regression analyses were performed to identify the risk factors predictive of URE. The factors included age (adjusted for all models), gender, and literacy. Multivariate logistic regression analyses were used to obtain the odds ratio and to identify the variables associated with URE and Fofo acceptance. The variables included age groups, gender, and educational status. $P < 0.05$ was considered significant. The odds ratio (OR) with the 95% Confidence Interval (CI) was calculated. All statistical analyses were performed using IBM SPSS (version 23.0; IBM Corp., Armonk, NY).

## Results

We screened 9,872 people (89% of the 11,085-target population) in the *Dongaria* community. The mean age of all screened people was 25.5 ± 18.8 years, and the mean age of the URE group was 47.1 ± 15.3 (range 8–74) years. We identified 744 people with refractive error in the cohort, and 76.8% (572/744) consented to refraction. Among those refracted, 16.4% (n = 94) consented to receive an on-site FoFo test, 61.2% (n = 350) were refracted at the VC, and 22.4% (n = 128) at the SC. The prevalence of refractive error was 7.5% in the screened people.

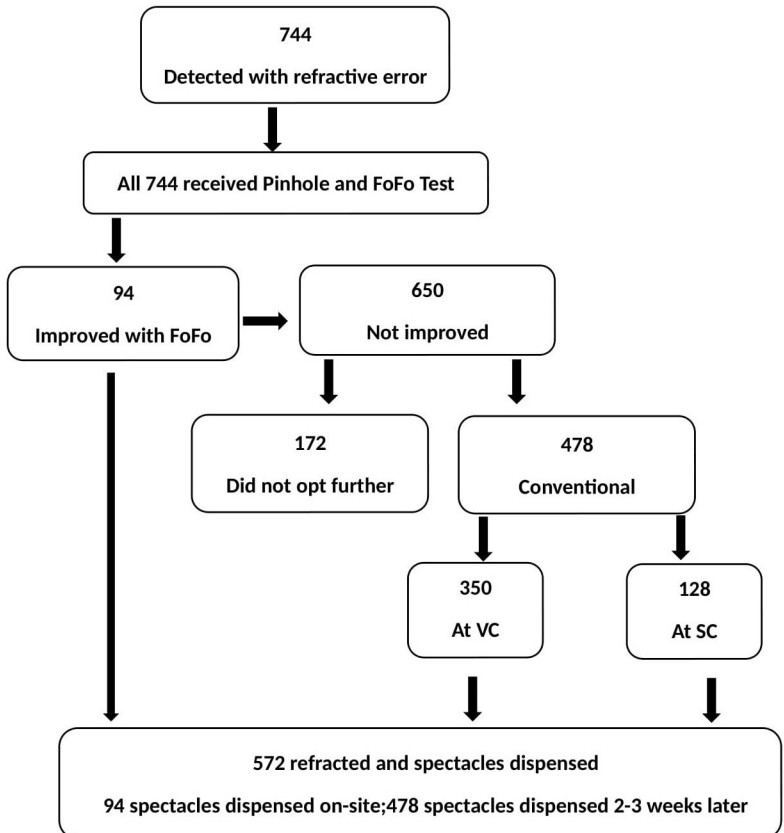

**Fig 2. Refraction and dispensing of spectacles in the Dongria PVTG community.**

Refractive error was more often detected in *Dongaria* females (63% vs 37%; Table 1). Spectacles were provided free of cost to all 572 people with refractive error.

Refractive error was low at 0.6% (n = 5) in children under 10 and high at 28.3% (n = 211) in 50–59 age group (Table 2). The range of refractive error was between +3.00 to -3.00 D. A prescription for spectacles was given to all individuals who recorded distance visual acuity better than 6/12 after the FoFo test.

**Table 1. Comparative demographic details of study participants and people with uncorrected refractive error.**

| Demographic Categories | | Screened population | People with RE |
|---|---|---|---|
| | | n = 9872 | n = 744 (7.5%) |
| | Mean age (years) | 25.5 ± 18.8 | 47.0 ± 15.3 |
| Gender | Male | 4481 (45.4%) | 274 (36.8%) |
| | Female | 5391 (54.6%) | 470 (63.2) |
| Literacy | Illiterate | 8515 (86.3%) | 647 (87.0%) |
| | Primary education | 968 (9.8%) | 75 (10.1%) |
| | Secondary education | 389 (3.9%) | 22 (2.9%) |
| Administrative block | Bissamcuttack | 4292 (43.5%) | 401 (53.9%) |
| | Muniguda | 3401 (34.4%) | 241 (32.4%) |
| | Kalyansinghpur | 2179 (22.1%) | 102 (13.7%) |

RE- Refractive error; Primary education (class 1 to 5); Secondary education (class 6 to 10).

**Table 2. Prevalence of refractive errors by age.**

| Age Group (Years) | Total Cohort | URE |
|---|---|---|
| | n (%) | n (%) |
| 0–9 | 2599 (26.3) | 5 (0.6) |
| 10–19 | 1798 (18.2) | 41(5.5) |
| 20–29 | 1889 (19.1) | 72 (9.6) |
| 30–39 | 1147 (11.6) | 90 (12) |
| 40–49 | 990 (10) | 158 (21.2) |
| 50–59 | 792 (8.02) | 211 (28.3) |
| 60 and above | 657 (6.6) | 167 (22.4) |

URE- Uncorrected refractive error.

We also noticed a gender variation in the incidence of refractive error in this community. More females had simple hyperopia (p = 0.04), and more males had compound myopic astigmatism (p = 0.01); Table 3.

Multiple logistic regression analysis showed older adults > 50 years of age (Adjusted OR: 49.8; 95% CI: 33–75; p <0.0001) and females (OR:1.5; 95% 95% CI: 1.2–1.7; p <0.0001) were at significantly higher risk of developing refractive error (Table 4).

Acceptance and accuracy of FoFo refraction were higher in young adults, females, those under 1D myopia, and literate individuals (Table 5).

## Discussion

The prevalence of uncorrected refractive error by the national survey in India is 29.6% in people under 50 and 13.4% in people ≥ 50 years [7]. Compliance with correcting spectacles depends on several factors–effective screening integrated with timely delivery, accurate refraction by appropriate health personnel (skilled optometrists/ VT), and using appropriate technology, including automated refraction where available, choice of a good range of spectacles frames, availability of low-cost spectacles and finally access to and continuity of comprehensive eye care [5, 8, 9]. Reports suggest higher compliance when dispensing the correcting spectacles soon after the eye screening [5, 10]. We used FoFo for automated refraction by the CHWs or VTs at the screening site to enable us to prescribe/dispense readily available spherical equivalent spectacles.

FoFo is a lightweight (32 grams), portable mechanical device for autorefraction. It uses two paper tubes and lenses (Fig 1A and 1B) and measures the spherical equivalent of the refractive error. Its advantages and disadvantages are shown in Table 6. Holding it at eye level in the non-dominant hand and using the dominant hand to slide the outer tube to find the point where the target appears sharp gives the spherical equivalent of URE. The user is expected to see through the FoFo and fix at a distant target (3 meters). The endpoint for each eye is obtained, and the SE is obtained from the scale set on FoFo. FoFo measures refractive error from + 4.00 to -7.00 D. FoFo relies on a subjective assessment of refractive error. It is dependent on the literacy level of the subject. In this first field trial of FoFo, the acceptance in the *Dongaria* community was 12.5%; it was not surprising, given the poor literacy in the community (13.7%). Furthermore, only 0.25% of *Dongaria* wore spectacles at the screening time. Onsite traditional retinoscopy poses multiple challenges–transport of equipment to the field, the need for alternative power owing to lack of electricity, and the unusual lighting conditions in this unique environment. Many people also hesitate to visit a VC or SC for refraction. All the

**Table 3. Prevalence of refractive errors by gender.**

| Refractive Error | Total | Male | Female | P value |
|---|---|---|---|---|
| | (n = 889 eyes) | (n = 310 eyes) | (579 eyes) | |
| Simple Hyperopia | 207 | 60(19.3%) | 147(25.3%) | 0.04* |
| Simple Myopia | 160 | 59(19%) | 101(17.4%) | 0.55 |
| Simple Hyperopic Astigmatism | 52 | 15(4.8%) | 37(6.3%) | 0.34 |
| Simple Myopic astigmatism | 110 | 37(11.9%) | 73(12.6%) | 0.77 |
| Compound hyperopic astigmatism | 99 | 33(10.6%) | 66(11.3%) | 0.73 |
| Compound myopic astigmatism | 202 | 85(27.4%) | 117(20.2%) | 0.01* |
| Mixed Astigmatism | 59 | 21(6.7%) | 38(6.5%) | 0.90 |

*Significant.

above factors made it challenging to compare FoFo results with the gold standard objective and subjective retinoscopy.

The current TOES-PVTG study documented URE among marginalised *Dongaria* tribals for the first time. The prevalence of URE was 7.5%, higher in adults over 30 but less than the reported prevalence of 10.2% In India [11]. The prevalence of simple myopia and hyperopia was higher in people 40–59 year group, similar to the reports of another population-based study in the adjoining state of Andhra Pradesh, India [12]. Important findings of this study concerning refractive error were (1) Females were more likely to have refractive error than males (1.5x; p<0.0001); (2) Hyperopia was more prevalent in *Dongaria* females; (3) Myopic astigmatism was more prevalent in males; (4) Refractive error was less common in children; (5) Lower order (+3D to -3D) refractive error was more prevalent in the adults.

We do not know the precise reasons for these findings. A markedly low prevalence of refractive error in the *Dongaria* children (0.6%) than reported in similar studies was very striking [13, 14]. It could be related to significant outdoor time and less exposure to near-devices (such as smartphones), both known to prevent the onset and progression of myopia [15]. We also attribute lower-order refractive error in adults to longer working outdoors in the fields.

FoFo was used in mass eye screening for the first time in India (Medline search on June 17, 2022). FoFo is easier to use than the liquid lens [16, 17], but the estimation of astigmatism may be superior in the liquid lens. Given the poor literacy in the *Dongaria* community, its acceptance was, in general, low (12.5%) but higher in young adults (OR: 23.4; 95% CI: 7.3–74.9; p

**Table 4. Multiple logistic regression for the association of uncorrected refractive errors with sociodemographic variables.**

| Characteristics | | Uncorrected refractive error | | FoFo acceptance | |
|---|---|---|---|---|---|
| | | Odds ratio | P value | Odds ratio | P value |
| | | (95% CI) for URE | | (95% CI) for URE | |
| Age (years) | 6–16 | Reference | | Reference | |
| | 17–30 | 4.9(3.1–7.7) | <0.0001 | 23.4(7.3–74.9) | <0.0001 |
| | 31–50 | 18.5(12.2–28) | <0.0001 | 19.7(6–64) | <0.0001 |
| | ≥ 51 | 49.8(33–75) | <0.0001 | 1.6 | 0.5 |
| Gender | Male | Reference | | Reference | |
| | Female | 1.5(1.2–1.7) | <0.0001 | 0.9(0.5–1.4) | 0.7 |
| Education | Illiterate | Reference | | Reference | |
| | Literate | 0.9(0.7–1.1) | 0.5 | 29.8(17–52.2) | <0.0001 |

CI- Confidence interval; URE- Uncorrected refractive error.

**Table 5. Comparison between FoFo accepted and not accepted (conventional) refraction.**

| Characteristics | FoFo accepted | FoFo not accepted (Conventional refraction) | P |
|---|---|---|---|
| | (n = 94) | (n = 478) | |
| Mean age (years) | 33.2 ± 11.2 | 50±14.7 | <0.0001 |
| Female gender | 71.2% (n = 67) | 63.3% (n = 303) | <0.0001 |
| Mean RE (D) | -0.3 ± 1 | -0.4 ± 1.7 | 0.9 |
| Mean myopic RE (D) | -0.9 ± 0.5 | -1.8 ± 0.2 | <0.0001 |
| Mean hyperopic RE (D) | 0.9 ± 0.3 | 0.9 ± 0.06 | 0.3 |
| Literacy | 76.5% (n = 72) | 4% (n = 20) | <0.0001 |

FoFo- Folding Phoropter; RE- Refractive error.

<0.0001) and literate subjects (OR:29.8; 95% CI: 17–52.2; p <0.0001). But the most significant benefit was an estimated savings of INR 137,000 (USD 1,611) towards the saved travel cost and wage loss of 94 individuals. FoFo costs INR 100 (nearly USD 1.2); it is significantly less expensive than other currently available commercially similar devices. Therefore, Fofo holds promise as a field tool in screening and autorefraction within its technological limitations.

The study's primary limitation was the absence of a gold standard–optometrists did not examine all patients using traditional objective and subjective refraction. Though desirable, doing conventional refraction would have been difficult. While we admit that a prescription spectacle could have been better, we have earlier reported that spherical equivalent spectacles up to 1D of the refractive error have satisfactory acceptance in rural India [18].

The study's major strength was that it was the first population-based refractive error evaluation in the *Dongaria* PVTG community and the use of the WHO-recommended IPEC (Integrated people-centred eye care) from the community to the secondary level of care. The Dongaria community is 8% (11,085 of 138,125) of the PVTG in Odisha. In this study, we could examine 89% of them and nearly all of them for the first time in their life. This is substantial in terms of social impact.

## Conclusion

Refractive error in the *Dongaria* community is not high, but for some peculiar characteristics. People do not use spectacles because of poor access, advocacy, and literacy. Until the general health-seeking behaviour improves and fixed eye care facilities, at least primary care, are established, one could continue with this mass eye screening and spectacles dispensing model. The productivity of females weaving the traditional shawls would also increase, bringing economic prosperity. The benefitting females would be the ambassadors for routine eye examinations and spectacles when needed. Other missing elements include inadequate human resources for eye health, motorable roads (difficult access to available health facilities), health facility linkage,

**Table 6. Advantages and disadvantages of FoFo.**

| Advantages | Disadvantages |
|---|---|
| Easy to understand and perform | Difficult for older people and children to understand |
| Requires less time | If performed improperly, the findings are erroneous. |
| Inexpensive | Cannot measure astigmatism |
| Lightweight | Fragile |
| Portable | Cannot determine higher refractive error |
| Does not require electricity or a battery | - |

and technologically improved but less expensive good quality refraction devices. However, there is also a fear that with increasing urbanisation, this secluded indigenous community could be exposed to electronic near-vision devices that would upset the current low refractive error in children and adults. We will need to figure out where to put a limit.

## Supporting information

**S1 Checklist. Additional information regarding the ethical, cultural, and scientific considerations specific to inclusivity in global research is included.**
(DOCX)

## Acknowledgments

The authors thank all the study participants, the community health workers, and the vision technicians who played a pivotal role in the screening process. The authors would also like to thank Mr Manav Jalan and Mr Ramanandan Mishro for planning and coordinating the project.

## Author Contributions

**Conceptualization:** Taraprasad Das, Akhila Bihari Ota, Suryasnata Rath.

**Data curation:** Debananda Padhy.

**Formal analysis:** Debananda Padhy, Suryasnata Rath.

**Investigation:** Debananda Padhy, Komal Avhad, Suryasnata Rath.

**Methodology:** Debasmita Majhi, Suryasnata Rath.

**Project administration:** Debasmita Majhi, Prachi Parimita Rout.

**Resources:** Akhila Bihari Ota, Suryasnata Rath.

**Supervision:** Rohit C. Khanna.

**Writing – original draft:** Debananda Padhy.

**Writing – review & editing:** Suryasnata Rath.

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
