## [Decision Letter · Decision Letter 0]

19 Jan 2023

PONE-D-22-33033Prevalence of Refractive error among Dongarias and Use of Folding Phoropter (FoFo) as a Field Device enabling Implementation Research in this Indigenous Community. Tribal Odisha Eye Disease Study (TOES) Report # 13PLOS ONE

Dear Dr. Rath,

Thank you for submitting your manuscript to PLOS ONE. After careful consideration, we feel that it has merit but does not fully meet PLOS ONE’s publication criteria as it currently stands. Therefore, we invite you to submit a revised version of the manuscript that addresses the points raised during the review process.

We look forward to receiving your revised manuscript.

Kind regards,

Godwin Ovenseri-Ogbomo, OD, MPH, PhD

Academic Editor

PLOS ONE

Journal Requirements:

3. You indicated that you had ethical approval for your study. Please clarify whether minors (participants under the age of 18 years) were included in this study. If yes, in your Methods section, please ensure you have also stated whether you obtained consent from parents or guardians of the minors included in the study or whether the research ethics committee or IRB specifically waived the need for their consent.

4. Please include a complete copy of PLOS’ questionnaire on inclusivity in global research in your revised manuscript. Our policy for research in this area aims to improve transparency in the reporting of research performed outside of researchers’ own country or community. The policy applies to researchers who have travelled to a different country to conduct research, research with Indigenous populations or their lands, and research on cultural artefacts. The questionnaire can also be requested at the journal’s discretion for any other submissions, even if these conditions are not met.  Please find more information on the policy and a link to download a blank copy of the questionnaire here: https://journals.plos.org/plosone/s/best-practices-in-research-reporting. Please upload a completed version of your questionnaire as Supporting Information when you resubmit your manuscript.

6. Thank you for stating the following financial disclosure: 

"Hyderabad Eye Research Foundation, L V Prasad Eye Institute, Hyderabad, India; Ministry of Tribal Affairs, Government of India; SC&ST Research Training Institute, Bhubaneswar, India; Naraindas Morbai Budhrani Trust, Mumbai, India."

7. In your Data Availability statement, you have not specified where the minimal data set underlying the results described in your manuscript can be found. PLOS defines a study's minimal data set as the underlying data used to reach the conclusions drawn in the manuscript and any additional data required to replicate the reported study findings in their entirety. All PLOS journals require that the minimal data set be made fully available. For more information about our data policy, please see http://journals.plos.org/plosone/s/data-availability.

8. PLOS requires an ORCID iD for the corresponding author in Editorial Manager on papers submitted after December 6th, 2016. Please ensure that you have an ORCID iD and that it is validated in Editorial Manager. To do this, go to ‘Update my Information’ (in the upper left-hand corner of the main menu), and click on the Fetch/Validate link next to the ORCID field. This will take you to the ORCID site and allow you to create a new iD or authenticate a pre-existing iD in Editorial Manager. Please see the following video for instructions on linking an ORCID iD to your Editorial Manager account: https://www.youtube.com/watch?v=_xcclfuvtxQ

Additional Editor Comments:

This is quite an innovative study that has the additional benefit of providing service to underserved indigenous population.

The authors indicated in the methods section (ln 120 - 125) that verbal consent was obtained from study participants because the Kui language has no written form. How than was written consent obtained at the secondary cites. Please clarify this. Under statistical analysis, (line 137): The prevalence of URE, the mean, median and range were calculated. It is not clear what mean, median and range were calculated. Please clarify this.

Reviewers' comments:

Reviewer's Responses to Questions

**Comments to the Author**

1. Is the manuscript technically sound, and do the data support the conclusions?

Reviewer #1: Yes

2. Has the statistical analysis been performed appropriately and rigorously? 

Reviewer #1: Yes

3. Have the authors made all data underlying the findings in their manuscript fully available?

Reviewer #1: Yes

4. Is the manuscript presented in an intelligible fashion and written in standard English?

Reviewer #1: Yes

5. Review Comments to the Author

Reviewer #1: PONE-D-22-33033 Prevalence of Refractive error among Dongarias and Use of Folding Phoropter (FoFo) as a Field Device enabling Implementation Research in this Indigenous Community.Tribal Odisha Eye Disease Study (TOES) Report # 13

General comments:

This paper is based on a very interesting project carried out in a remote tribal area of India amongst a population with very special health care, demographic, and socio-economic characteristics that makes typical scientific study and reporting difficult or perhaps impossible. Thus, the methods used in this project are by necessity different than what typically gets studied and reported in the medical literature. This poses a number of difficulties that the authors do a great job in mitigating in order to provide important and very useful information to the audience. They were forced to do things in a manner that perhaps would otherwise not be acceptable in a more typical study, but in the context of the community they were working with, nonetheless valuable and most importantly, really interesting and creative. It also necessitates my reviewing this paper in a different manner than I typically do. I chose to do editing of the document that was submitted within the document itself and add additional comments where I believe indicated and useful to the authors. My edits are in both italics and a yellow colored font.

I am certain that this paper deserves publication in PLOS ONE with some additional editing by the authors. Given some allowances for the points above, this paper is different in some ways than a “typical” publishable research paper. However, it is perhaps particularly appropriate for a journal like PLOS ONE to give an opportunity to these authors that might be less acceptable to more “traditional” journals.

The authors do point out the many limitations created by the population and environment this study was conducted in. It might nonetheless benefit by some expansion of those limitations, as I suggest within my comments and edits within the paper. I have made an effort throughout the paper to make edits on language, syntax, and usage, which the authors may take or leave, but I think help make it a better and clearer paper.

Beyond the general comments above, a few more suggestions. First, you need a better description of the FoFo within this paper, and not just in references. A photo or diagram of the device, and maybe a photo of a subject undergoing the self-refraction process with the FoFo, would make clear to the reader what you are actually doing.

A better description of your definitions and categorizations of refractive error numerically in or adjacent to the refractive error table would be useful.

How are you defining literacy, or for that matter, non-literacy?

I was not clear how you calculated or used the term odds ratio in a later table. Perhaps define / clarify that better.

Finally, at the end of the paper, you come across a bit too speculatively for why the refractive errors are what this population has in comparison to other populations. Seems like a pretty far reach to me in some of your speculations. You might consider tempering things here.

This is a valuable contribution to the literature and with some mild revision, worthy of publication in PLOS ONE.

6. PLOS authors have the option to publish the peer review history of their article (what does this mean?). If published, this will include your full peer review and any attached files.

Reviewer #1: **Yes: **Bruce Moore OD

Professor Emeritus

New England College of Optometry

mooreb@neco.edu

---

## [Author Response · Author response to Decision Letter 0]

11 Mar 2023

1.How was written consent obtained at the secondary centre? 1.This is added. (Page no.7 and line 133-34).

2.Please clarify this: The prevalence of URE, the mean, median, and range were calculated using statistical analysis. What exactly was calculated for mean, median, and range is unclear. 

Response: Additional information are added. (Page no.8 line 153-54)

3.Better description of FoFo:

First, you need a better description of the FoFo within this paper, and not just in references. A photo or diagram of the device, and maybe a photo of a subject undergoing the self-refraction process with the FoFo, would make clear to the reader what you are actually doing.

Response: The FoFo is described, advantage-disadvantage tabulated and appropriate images inserted. (line 114-18 &141-43 & 211-29 and Figure 1 &Table 6 and page 6-8 & 14-15).

4.Description of definitions and categorizations of refractive error numerically:. It is added. (Page 7 and line 121-30).

5.Definition of Literacy:It is now added. (Page 8 and line 149-51).

6.Use of odds ratio: We have done it. (Page 8 and line 156-59)

7.Review of reference list: One reference was added to the discussion's opening line (Page 18 and line 297-98)

The reference # 7 is corrected. 

8. Finally, at the end of the paper, you come across a bit too speculatively for why the refractive errors are what this population has in comparison to other populations. Seems like a pretty far reach to me in some of your speculations. You might consider tempering things here.

Response: We thank the reviewer for the comment. We have deleted the following sentences - But we suppose, the higher prevalence of refractive error in females may possibly be due to their engagement in demanding near-tasks like shawl weaving. In general, a relatively higher prevalence of hypermetropia in females may be attributed to the reported shorter axial length, and decrease in anterior chamber depth and lens thickness in women than men. The higher prevalence of myopic astigmatism in males noted by us does not match other similar studies. We suspect it could be due to an increased involutional lid laxity;

---

## [Editor Report · Decision Letter 1]

10 Apr 2023

Prevalence of Refractive error among Dongarias and Use of Folding Phoropter (FoFo) as a Field Device enabling Implementation Research in this Indigenous Community. Tribal Odisha Eye Disease Study (TOES) Report # 13

PONE-D-22-33033R1

Dear Dr. Rath,

We’re pleased to inform you that your manuscript has been judged scientifically suitable for publication and will be formally accepted for publication once it meets all outstanding technical requirements.

Kind regards,

Godwin Ovenseri-Ogbomo, OD, MPH, PhD

Academic Editor

PLOS ONE

Additional Editor Comments (optional):

Thank you for addressing the points raised by the reviewer.
---

## [Editor Report · Acceptance letter]

28 Apr 2023

PONE-D-22-33033R1 

Prevalence of Refractive error among *Dongarias* and Use of Folding Phoropter (FoFo) as a Field Device enabling Implementation Research in this Indigenous Community. Tribal Odisha Eye Disease Study (TOES) Report # 13. 

Dear Dr. Rath:

I'm pleased to inform you that your manuscript has been deemed suitable for publication in PLOS ONE. Congratulations! Your manuscript is now with our production department. 

Kind regards, 

on behalf of

Dr. Godwin Ovenseri-Ogbomo 

Academic Editor

PLOS ONE